# Agri-Environment Atmospheric Real-Time Monitoring Technology Based on Drone and Light Scattering

**Yuan Liu, Xun He, Wanzhang Wang \*, Chenhui Zhu, Ruibo Jian and Jinfan Chen**

College of Mechanical and Electrical Engineering, Henan Agricultural University, Zhengzhou 450002, China
\* Correspondence: wangwz@henau.edu.cn

**Abstract:** The emission of particulate matter (PM) from agricultural activities, such as concentrated animal feeding, straw combustion, and mechanized harvest, is a hot issue in the sustainable development of agriculture, which has attracted more and more attention from government departments and researchers. However, the research on the transport of particulate matter in the agri-environment still lacks flexible and efficient measurement methods to obtain real-time and accurate spatial distribution data. The objective of our study is to produce a new intelligent platform for agri-environment atmospheric monitoring with high mobility, temporal and spatial resolution, and remote data transmission function to overcome the shortcomings of traditional atmospheric particulate matter monitoring stations, such as small particle size range, immovability, and high cost. Through the light scattering sensor, microcontroller, and wireless data transmission device assembled on the high-mobility drone, the platform could measure the mass concentration of PM2.5, PM10, and TSP at different spatial points in the agri-environment and transmit the measurement data to the receiving device on the ground through three modes: CLOUD, TCP, and UDP. We also developed monitoring software based on the Android platform, which could complete the connection of device and real-time monitoring of measurement data on the ground. Compared with stationary measurement devices, the biggest advantage of our mobile monitoring system is that it has the ability to measure the concentration of TSP and the vertical distribution of PM, which is very important for the research of agricultural environmental particulate matter emission characteristics. After the sensor and communication performance experiments, the sensors had high consistency in the overall change trend, and the communication accuracy rate was high. We carried out a flight measurement comparison experiment at the Wenhua Road Campus of Henan Agricultural University, and the measurement data were highly consistent with the data from the national monitoring stations. We also conducted an agri-environmental atmospheric measurement experiment in Muzhai Village and obtained the vertical distribution data of PM concentration at the nearby measuring point when the harvester was working. The results showed that after the harvester worked for a period of time, the PM2.5, PM10, and TSP concentrations reached the maximum at the altitude of 20 m at the measurement point, which were 80, 198, and 384 $\mu g/m^3$, respectively, 2.64~3.10 times the particle concentration in the environment before the harvester began to work. Our new platform had high mobility, sensitive reading, and stable communication during the experiment, and had high application value in agricultural environmental monitoring.

**Keywords:** agri-environmental monitoring; particulate emission; temporal and spatial distribution; drone; long-distance wireless communication



## 1. Introduction

Environmental pollution has endangered planetary health [1] since the Industrial Revolution in the late 18th century. To achieve a world that is healthy for both the global biosphere and human civilization, the critical challenge is to curb the environmental impact of socioeconomic activities. Among the environmental problems affecting human health, the biggest threat is the inhalation of particulate matter with an aerodynamic diameter of

less than or equal to 2.5 μm, abbreviated to PM2.5 [2]. As an important component of air pollution, atmospheric particulate matter (APM) has attracted worldwide attention [3–5]. APM is a general term for various solid and liquid suspended particulate substances in the atmosphere [6], which are uniformly dispersed in the air to form a relatively stable suspension system [7]. Several studies have demonstrated the effects of PM2.5 on human health [8,9]. Higher PM2.5 concentrations not only increase the risk of respiratory diseases such as asthma and bronchitis, but also increase the risk of cardiovascular disease. Worldwide exposure to ambient PM2.5 causes over 4 million premature deaths annually, with most of these deaths occurring in developing countries [10].

There is also a serious problem of particulate matter emissions in the agricultural environment, which has received great attention in China. For example, the particulate matter generated from soil erosion by wind and mechanized harvesting is mainly soil dust particles and also contains some fungi and other microorganisms [11,12]. The reason for the generation of these particles is the disturbance of wind and agricultural equipment to the crop plants and soil. Affected by many factors, different regions, different soil types, and different soil depths cause different soil particle size distributions [13], and even under different tillage methods, soil particle size distributions will also change [14]. Since particles larger than 100 μm in the air will sediment quickly under the action of gravity, the general research object is particulate matter with an aerodynamic diameter of less than or equal to 100 μm, abbreviated to PM100, which is also called total suspended particles (TSP) because they can be suspended in the air for a long time. According to the soil analysis method of Gee and Bauder [15], taking cultivated land as an example, among all soil particles with a particle size of less than 100 μm, the particles with a particle size between 20 and 50 μm accounted for 20.41%, and the particle size between 50 and 100 μm accounted for 78.9%. It can be seen that the particulate matter produced by agricultural production activities is mostly large particles, which is very different from the common research that mainly uses PM2.5 and PM10 as the measurement objects. Although the sedimentation rate of these large particles is higher than that of PM2.5 and PM10, on a local scale, the accumulation of high concentrations of TSP will still cause harm to the health of agricultural personnel, production safety, and the surrounding environment. On the one hand, the particulate matter emitted from agricultural activities will diffuse to the surrounding villages and affect the health of the population in rural areas. On the other hand, under the action of wind, this particulate matter will be transmitted over a long distance to reach urban areas. In the process of transmission, the particulate matter will be deposited and affect the local ecosystem and population health. With the concern about the pollution caused by atmospheric particles, government departments have also taken some actions. Through these actions, especially the actions taken under the UNECE Convention on Long-Range Transboundary Air Pollution, air emissions were substantially reduced, and ecosystem impacts decreased. Widespread scientific research, long-term monitoring, and integrated assessment modelling formed the basis for the policy agreements [16].

For real-time monitoring, the government has built some fixed monitoring sites, which can monitor the mean value of PM2.5 and PM10 in a specific time interval (for example, 1 h) and upload it to the online website for publication. However, due to cost reasons, traditional fixed monitoring sites are generally set far apart (usually more than a few kilometers or even tens of kilometers), are mostly located in urban areas such as schools, parks, and residential areas, and rarely involve agricultural areas. For the study of particulate matter emissions and hazards in agricultural activities, it is very important to obtain measurement data with high spatial and temporal resolution through flexible and mobile monitoring platforms. However, the measurement instruments used in air quality monitoring stations generally use micro-oscillating balances and beta-ray attenuation methods. These instruments present difficulty when moving because they are large and heavy. Due to the influence of complex meteorological conditions, the spatial distribution of particles produced by agricultural activities is very complex, and it is difficult to obtain complete data by fixed measurement devices. Moreover, the expensive equipment price has

become the bottleneck of wide application. With the development of drone and remote data transmission technology [17], some flexible and efficient intelligent monitoring systems are also gradually applied in agricultural production. Therefore, how to use highly flexible drones to carry a low-cost and lightweight measurement system is very important for the study of agricultural environmental particulate matter emission characteristics.

The research background of this paper is shown in Figure 1.

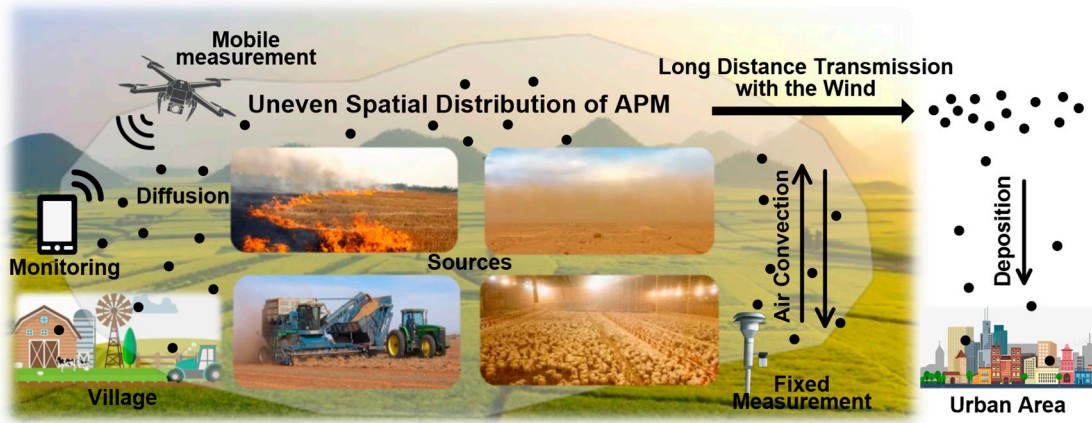

**Figure 1.** Diffusion, transmission, and measurement of particulate matter from agricultural activities.

The objective of our study is to produce a sensor fusion system with high spatial and temporal resolution and remote data transmission function, which is assembled on a highly maneuverable drone platform and can not only measure PM2.5 and PM10, but also TSP and other environmental information such as temperature and humidity. Among this information, the concentration of TSP is a very important indicator in the agricultural environment, but it has not been considered in the previous measurement systems and national monitoring stations that focus on urban environments. Our system can transmit the measurement data to the receiving device on the ground for real-time monitoring. This study overcomes the shortcomings of traditional atmospheric particulate matter monitoring stations, such as small particle size range, immovability, and high cost, and provides more abundant data for the follow-up research on particulate matter emissions and hazards in agricultural activities.

## 2. Related Studies

### 2.1. Drone Technology

In recent years, drone technology has been widely used in agricultural production. For example, drone remote sensing technology is used to monitor irrigation and crop growth, and drones for plant protection are used to spray pesticides. Drone products on the market have high stability, but while providing certain convenience, their expensive price and extremely limited scalability also bring great restrictions. Therefore, an assembling drone is the best choice. The flight control system is the core component for the drone to complete the whole flight process, such as takeoff, air flight, mission execution, and return, and plays a decisive role in the maneuverability and stability of the drone. The flight control system calculates the attitude data returned by the gyroscope and completes the action and flight attitude adjustment by changing the speed of propellers according to the remote-control command.

The Earth-centered, Earth-fixed coordinate system (acronym ECEF) and body coordinate system are established, as shown in Figure 2. The propellers are numbered clockwise. In the figure, $\theta$ represents the pitch angle, $\varphi$ the roll angle, $\psi$ the yaw angle, and the red arrow represents the rotation direction of the propeller.

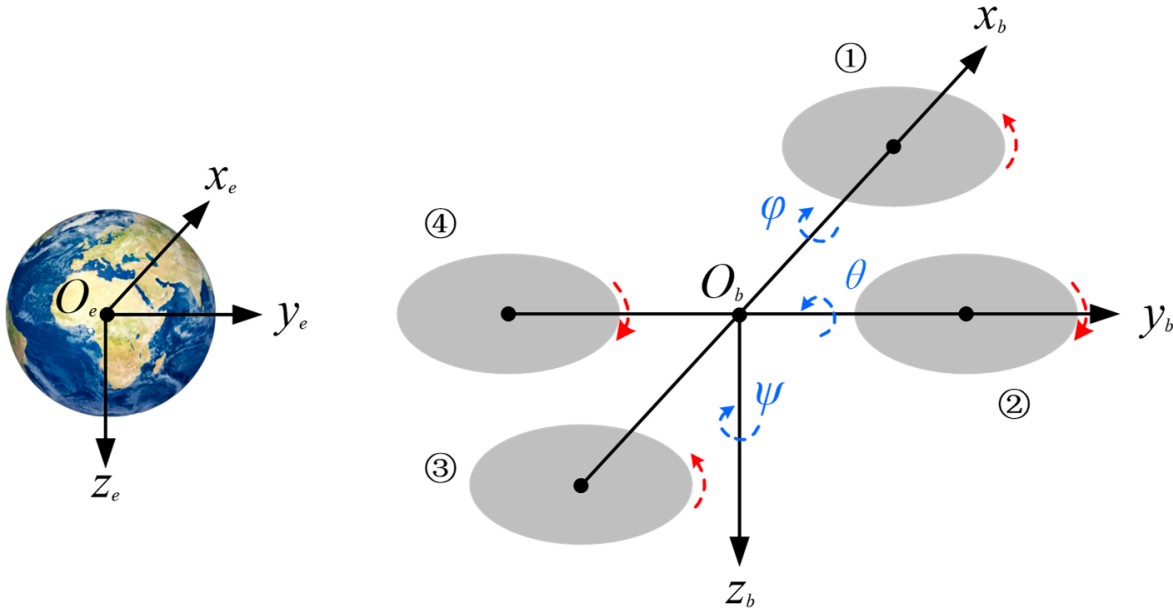

**Figure 2.** The Earth-centered, Earth-fixed coordinate system and body coordinate system: ①–④ are propeller numbers, where ① and ③ are in the same direction, and ② and ④ are in the other direction.

The rotation matrix of any vector from the body coordinate system $O_b x_b y_b z_b$ to ECEF $O_e x_e y_e z_e$ is expressed as the following:

$$R_b^e = \begin{bmatrix} \cos\theta\cos\psi & \cos\psi\sin\theta\sin\varphi - \sin\psi\cos\varphi & \cos\psi\sin\theta\cos\varphi + \sin\psi\sin\varphi \\ \cos\theta\sin\psi & \sin\psi\sin\theta\sin\varphi + \cos\psi\cos\varphi & \sin\psi\sin\theta\cos\varphi - \cos\psi\sin\varphi \\ -\sin\theta & \sin\varphi\cos\theta & \cos\varphi\cos\theta \end{bmatrix} \quad (1)$$

Let $U_1$, $U_2$, $U_3$, and $U_4$ denote the lift, roll moment, pitching moment, and yawing moment, which are linearly related to the square of the propeller angular velocity $\omega_i$, the angular velocity of the propeller numbered $i$, the total moment of inertia of the entire motor rotor and propeller $J_{RP}$, and $\Omega = -\omega_1 + \omega_2 - \omega_3 + \omega_4$. The six-degree-of-freedom control model of a quadrotor can be described by the following system of equations:

$$\begin{cases} \ddot{x} = -\frac{U_1}{m}(\cos\psi\sin\theta\cos\varphi + \sin\psi\sin\varphi) \\ \ddot{y} = -\frac{U_1}{m}(\sin\psi\sin\theta\cos\varphi - \cos\psi\sin\varphi) \\ \ddot{z} = g - \frac{U_1}{m}\cos\varphi\cos\theta \\ \ddot{\varphi} = \frac{1}{I_{xx}}[U_2 + qr(I_{yy} - I_{zz}) - J_{RP}q\Omega] \\ \ddot{\theta} = \frac{1}{I_{yy}}[U_3 + pr(I_{zz} - I_{xx}) - J_{RP}p\Omega] \\ \ddot{\psi} = \frac{1}{I_{zz}}[U_4 + pq(I_{xx} - I_{yy})] \end{cases} \quad (2)$$

where $m$ is the mass of the drone and $g$ the acceleration of gravity, and the control information $U$, moment of inertia matrix $I$, and angular velocity $\omega$ of the drone body are expressed as follows:

$$U = \begin{bmatrix} U_1 \\ U_2 \\ U_3 \\ U_4 \end{bmatrix} = \begin{bmatrix} k_L & k_L & k_L & k_L \\ 0 & -k_L l & 0 & k_L l \\ k_L l & 0 & -k_L l & 0 \\ k_M & -k_M & k_M & -k_M \end{bmatrix} \begin{bmatrix} \omega_1^2 \\ \omega_2^2 \\ \omega_3^2 \\ \omega_4^2 \end{bmatrix} \quad (3)$$

$$I = \begin{bmatrix} I_{xx} & & \\ & I_{yy} & \\ & & I_{zz} \end{bmatrix}, \quad \omega = \begin{bmatrix} p \\ q \\ r \end{bmatrix} = \begin{bmatrix} \dot{\varphi} \\ \dot{\theta} \\ \dot{\psi} \end{bmatrix} \quad (4)$$

where $k_l$ is the lift coefficient, $k_M$ is the torque coefficient, and $l$ is the distance from the propeller axis to the drone's center of mass. It can be seen from Equations (2) and (3) that

the control of the drone's position and flight attitude can be achieved by changing the rotational speed of different propellers.

### 2.2. Light Scattering Particle Measurement Technology

With the increasing awareness of air pollution prevention and control [18–20], many lightweight, low-cost particulate matter sensors are produced by manufacturers, such as DSM501A and GP2Y1010AU0F particle sensors launched by Syhitech (Seoul, Korea) and Sharp (Osaka, Japan) [21], PMS5003T particle sensors launched by Plantower (Nchang, China) [22], and palm-sized optical particle sensors launched by Panasonic (Osaka, Japan) [23]. Thanks to the rapid technological evolution of drones, sensors, wireless communication, and tiny onboard computers, it is possible to realize the mobile atmospheric particulate matter measurement of the agri-environment with high spatial and temporal resolution under better technical support.

Light scattering particle sensors are based on Mie theory [24]. These sensors draw the gas to be tested into the detection dark room through a fan and use a stable light source to illuminate the particles of different diameters to generate scattered light with different intensities. The intensity of the light scattering of a single particle based on the Mie theory is described in the following equation:

$$I_s = \frac{\lambda^2 I_0}{4\pi R^2}\left(|S_1|^2 \sin^2 \varphi + |S_2|^2 \cos^2 \varphi\right) \tag{5}$$

where $\lambda$ is the wavelength of incidence light, $I_0$ the intensity, $R$ the observation distance, $\theta$ the angle between the light path and the scattered light, $S_i$ the amplitude function related to $\theta$, and $\varphi$ stands for the angle between the vibration plane of incident light and scattering surface. The scattered light was converted into pulse signal with different amplitudes through the photoelectric conversion device, so as to obtain the mass concentration of particles. The working principle is shown in Figure 3.

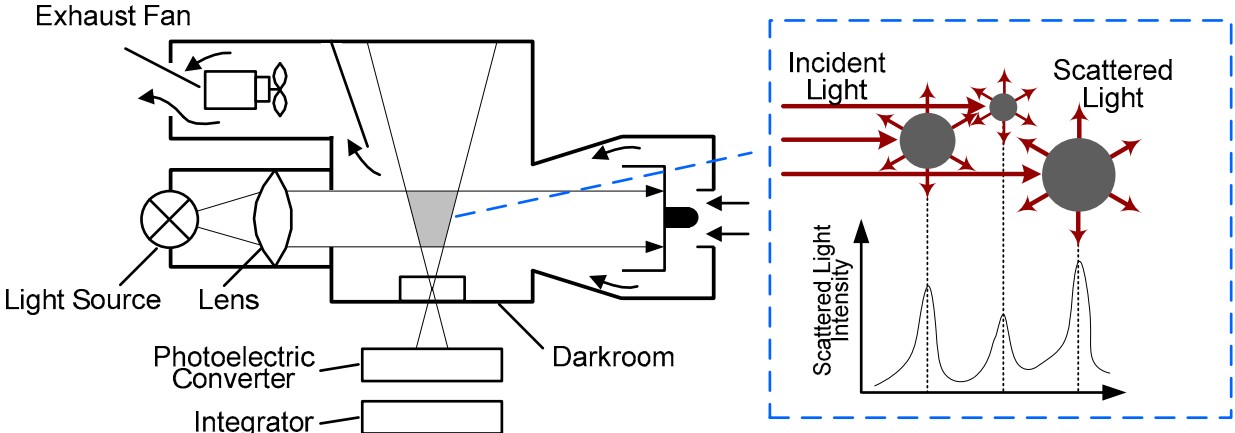

**Figure 3.** The working principle of light scattering particle sensor.

## 3. Materials and Methods

### 3.1. Overall System Architecture

Figure 4 depicts the overall architecture of our proposed monitoring platform. The platform we designed comprises four subsystems: flight power and control subsystem (FPCS), multi-sensor fusion subsystem (MSFS), long-distance wireless communication subsystem (LWCS), and ground real-time monitoring subsystem (GRMS).

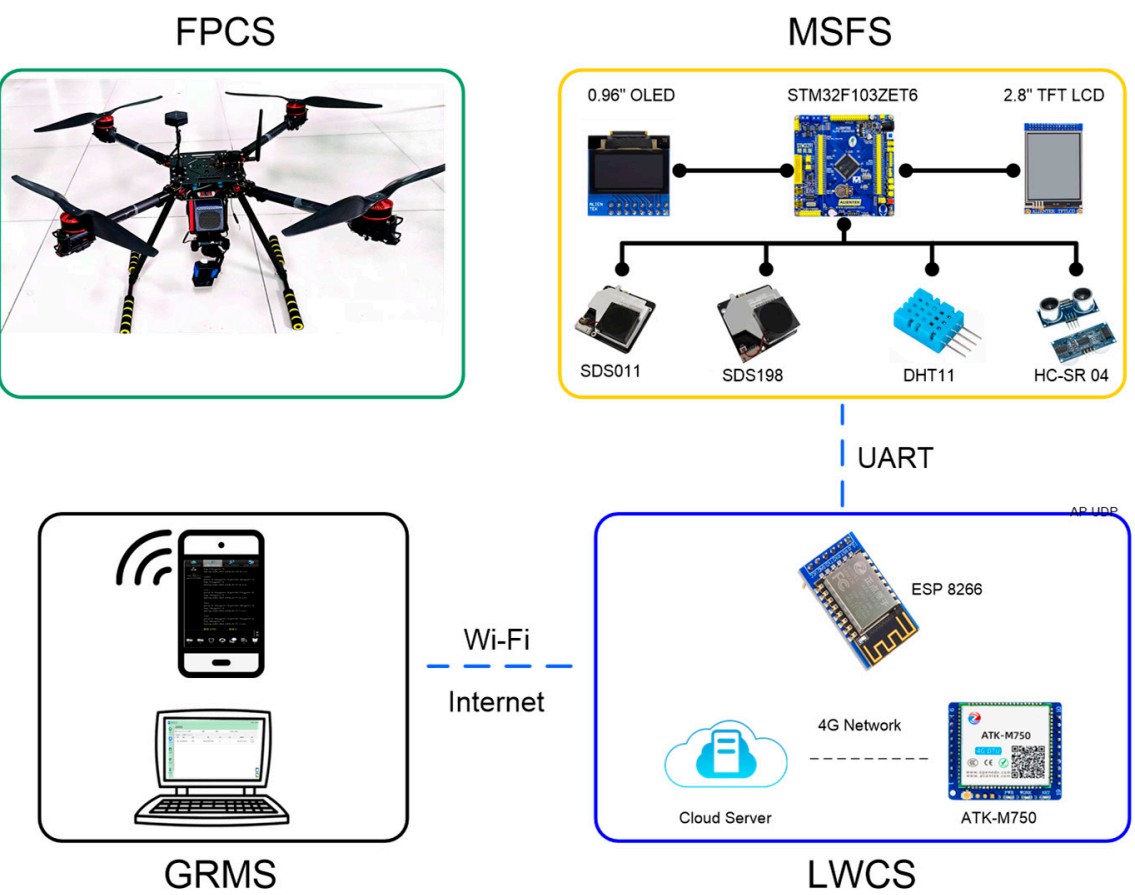

**Figure 4.** Overall architecture of agri-environment atmospheric monitoring platform.

The flight power and control subsystem is composed of a 4-rotor drone using Pixhawk open source flight controller and a ground flight remote controller. The diagonal distance of the drone is 680 mm, and it has GPS and aerial photography functions. Its excellent size, load capacity, and mobility can meet the installation and work requirements of other subsystems.

The multi-sensor fusion subsystem consists of two light scattering sensors, a temperature and humidity sensor, an ultrasonic distance sensor, a microcontroller, and other auxiliary devices. The subsystem can measure the mass concentration data of PM2.5, PM10, and TSP, and obtain agri-environmental data such as temperature and humidity. Through the ultrasonic sensor, some distance information can be obtained.

The long-distance wireless communication subsystem consists of a WI-FI module, a 4G-DTU module, and a cloud server. We designed 3 data transmission modes to ensure timely and reliable delivery of measurement data to cloud servers or mobile phones. The ground real-time monitoring subsystem is responsible for receiving and displaying detection data from the cloud server or directly from the WI-FI module.

The following describes the details of the FPCS, MSFS, LWCS, and GRMS development processes.

### 3.2. Flight Power and Control Subsystem

FPCS comprises a 4-rotor drone, 8000 mAh 6S battery, GPS, Pixhawk flight control system, remote controller, aerial camera, video transmission module, ground receiver, etc. The main parameters and configurations are shown in Table 1.

**Table 1.** Parameters and devices for FPCS.

| Parameter/Device | Value/Model |
|---|---|
| Diagonal distance (propellers excluded) | 680 mm |
| Takeoff weight (battery included) | 2800 g |
| Max load | 3000 g |
| Max flight time (no wind) | 35 min (at a consistent 25 km/h) |
| Max flight distance (no wind) | 15 km (at a consistent 25 km/h) |
| Max Speed | 65 km/h |
| Max takeoff altitude | 2000 m |
| Transmission distance | 2000 m |
| Carbon fiber propeller | 381 mm (15 inches) |
| Battery | 6S 8000 mAh |
| Electric motor | D4310-400KV |
| Electronic speed control | HobbyWing 40A |
| GPS | RadioLink M8N |
| Flight control system | Pixhawk 2.4.8 |
| Remote controller | RadioLink AT9S Pro |
| Camera | SJCOM SJ4000 |
| Video transmission | TS832 |
| Ground receiver | Hawk Eye LP-School |

The main function of FPCS is to provide installation platforms and mobility capabilities [25,26] for subsystems such as MSFS and LWCS. The diagonal distance of the drone is relatively large, making it possible to install a variety of sensors and data transmission components. At the same time, the high-performance motor and large-capacity lithium battery can ensure that the monitoring system has high load capacity, mobility, and endurance during operation. The assembled FPCS system is shown in Figure 5.

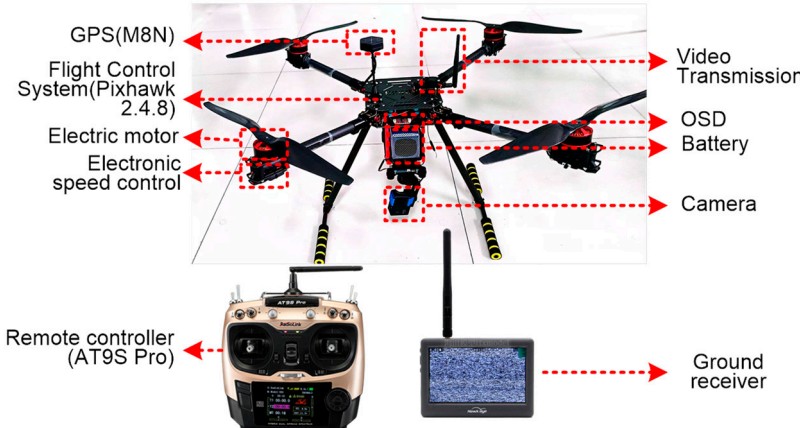

**Figure 5.** Assembled FPCS and part assignment.

*3.3. Multi-Sensor Fusion Subsystem*

3.3.1. Sensor Selection

For our platform, lightweight and high time resolution should be the primary criteria for sensor selection. The instruments of the micro-oscillation balance method and the β-ray method used in the traditional monitoring station are too large and are not suitable for our platform. In recent years, manufacturers have come forth with cost-effective sensors for measuring ambient and indoor particulate matter concentration [27,28]. Especially with the development of light scattering particle detection technology, more portable sensors are constantly appearing.

After investigation, we selected 4 common particulate matter sensors and compared them in terms of measurement objects, communication methods, product sizes, etc. The details of the comparison are shown in Table 2.

**Table 2.** Major specifications of four common PM sensors.

| Parameter | SDS011 | SDS198 | PMS5003 | GP2Y1014AU0F |
|---|---|---|---|---|
| Principle | Light scattering | Light scattering | Light scattering | Light scattering |
| Voltage | 5.0 V | 5.0 V | 5.0 V | 5.0 V |
| Communication | Digital signal | Digital signal | Digital signal | Analog signal |
| Particle diameter | PM2.5, PM10 | TSP (PM100) | PM1.0, PM2.5, PM10 | PM2.5 |
| Measuring range | 0~2000 µg/m$^3$ | 0~20,000 µg/m$^3$ | 0~1000 µg/m$^3$ | 0~500 µg/m$^3$ |
| Size (L × W × H) | 71 × 70 × 23 mm | 71 × 70 × 23 mm | 50 × 38 × 21 mm | 46 × 30 × 17 mm |
| Weight | 53 g | 49 g | 32 g | 15 g |

Considering the measurement requirements of particle diameter and concentration in agri-environments, we chose SDS011 and SDS198 by Shandong Nova Fitness Co., Ltd. (Jinan, China). The combination of this sensor can measure the concentration of particulate matter with three particle sizes: PM2.5, PM10, and TSP. These two models of particulate matter sensors have been calibrated at the factory and have been tested for comparison with the Model 8533 aerosol and dust monitor by TSI Instrument Co., Ltd. (Shoreview, MN, USA).

The DHT11 is a basic, ultra-low-cost digital temperature and humidity sensor. It uses a capacitive humidity sensor and a thermistor to measure the surrounding air and spits out a digital signal on the data pin (no analog input pins needed). The working voltage of DHT11 is 3.3~5.5 V, the temperature measurement range is −20~+60 °C, the measurement accuracy is ±2 °C, the relative humidity measurement range is 5~95%, and the measurement accuracy is ±5%. At its core, the HC-SR04 Ultrasonic distance sensor consists of two ultrasonic transducers. The one acts as a transmitter which converts electrical signal into 40 KHz ultrasonic sound pulses. The receiver listens for the transmitted pulses. If it receives them, it produces an output pulse whose width can be used to determine the distance the pulse travelled, which is twice the distance to be measured. The measured distance *d* (cm) can be obtained by the following equation:

$$d = \frac{v \times 10^2 \times t}{2 \times 10^6} = 0.017 \times t \tag{6}$$

where *t* is the output pulse width (us) from the sensor and *v* is 340 m/s, which is the speed of sound at 1 standard atmosphere pressure and an air temperature of 15 °C. At 1 standard atmosphere pressure, there is a certain relationship between the speed of sound and temperature, as shown in Equation (6):

$$v_T = 331 \times (1 + \sqrt{1 + \frac{T}{273}}) \tag{7}$$

where *T* is the temperature in Celsius and $v_T$ is the speed of sound at *T* degrees Celsius. After obtaining the temperature via DHT11 and combining the two equations, the temperature-corrected distance $d_c$ can be expressed by Equation (7):

$$d_c = 0.01655 \times (1 + \sqrt{1 + \frac{T}{273}}) \times t \tag{8}$$

Different installation methods can achieve different functions, such as avoiding obstacles, stabilizing height, measuring distance, etc. We chose STM32F103ZET6 as the MCU of the subsystem and used a 2.8″ TFT LCD Display and a 0.96″ OLED Display to show system status and measurement information. In addition, the subsystem also includes some basic components such as buttons and positioning boards.

### 3.3.2. Assembly of MSFS

Each part of the MSFS was mounted on a positioning plate with a size of 150 × 150 mm. The total MSFS weight is 361 g. We considered the assignment of each component with the center-of-gravity balance during assembly. The assembled subsystem is shown in Figure 6.

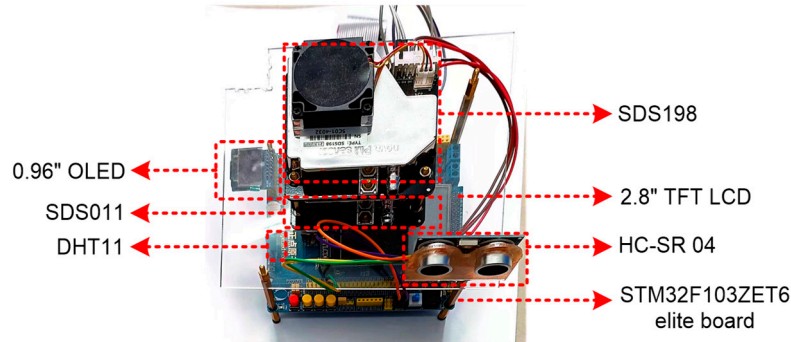

**Figure 6.** Assembled MSFS and part assignment.

After the system is powered on, MSFS first needs to set the working mode by pressing the button, so that the sensor can be initialized. Then, according to their respective communication protocols, the MCU sends them a dedicated data request command. The sensor starts to measure after receiving the command and stores the measured value in the protocol message or high-level digital signal of a specific length. The MCU analyzes these messages or signals, obtains measurement values, and transmits them to the remote wireless communication subsystem at the same time.

### 3.4. Long-Distance Wireless Communication Subsystem

According to the requirements of the monitoring system for the real-time measurement data, combined with the agricultural environment, we designed three data communication modes, namely 4G-CLOUD mode, AP-TCP mode, and AP-UDP mode. These three modes can ensure stable and reliable data transmission under different operating environments and transmission distances. We completed all functions of data transmission through the cloud server and two modules, ESP8266 and ATK-M750. Figure 7 depicts the LWCS connection architecture.

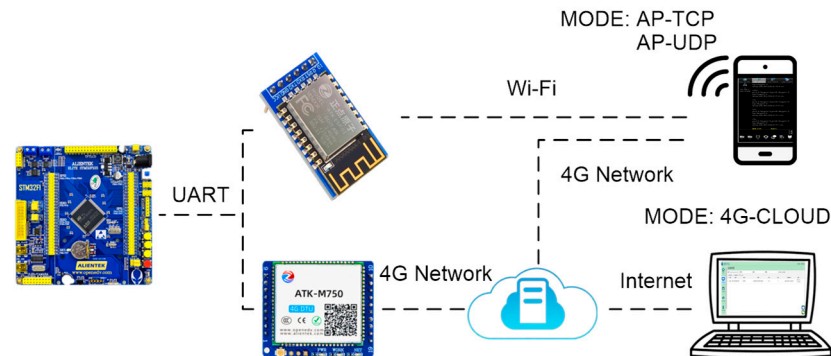

**Figure 7.** LWCS connection architecture.

### 3.4.1. The 4G-CLOUD Mode

The 4G-CLOUD mode is the most frequently used mode, because it is not limited by transmission distance and obstacles. Where there is a 4G network, data can be transmitted to the cloud server through the ATK-M750 module and a 4G sim card. The working process of 4G-CLOUD mode is as follows. First, the MCU sends AT commands through the serial port to perform self-check and initial configuration of the ATK-M750 module, and then

connects to the cloud server through the HTTP protocol (or MQTT) for authentication. When all preparations are completed, continuous data transfer can be achieved.

### 3.4.2. AP-TCP and AP-UDP Mode

Some remote agricultural areas may have no 4G network, so we designed another 2 modes for data communication. Both AP-TCP and AP-UDP modes are implemented through the ESP8266 module. AP-TCP establishes a TCP connection through a hotspot and forms a dedicated data transmission channel through the "handshake protocol" for continuous data transmission. AP-UDP mode does not need to establish a dedicated data transmission channel. It sends encapsulated and ordered data packets through hotspots, and the receiver receives these data packets and reorganizes them in order. The working process of these two communication modes is as follows. First, the MCU sends the corresponding AT command to the ESP8266 module through the serial port to enable the Wi-Fi hotspot and set the relevant parameters; secondly, the mobile terminal device connects to the Wi-Fi hotspot, obtains the IP address through the DHCP protocol, and establishes a TCP/UDP server; thirdly, the MCU sends the corresponding AT command to the ESP8266 module through the serial port and connects to the TCP server to realize continuous data transmission.

The M750 and ESP8266 modules are very light, meeting the lightweight requirements of our mobile platform. At the same time, compared with the existing schemes, they have lower costs, and do not need to set up a separate gateway.

### 3.5. Ground Real-Time Monitoring Subsystem

In order to receive the measurement values of the monitoring system in real time, we designed a variety of ground receiving schemes based on PC and Android mobile devices. We developed an APP based on the Android platform to receive measurement data in 4G-CLOUD mode, and also used the WEB browser on the PC side to access data in B/S mode. We also used the LAN communication software commonly used on the Android platform to receive data in AP-TCP and AP-UDP mode. Figure 8 shows the data receiving interface of these ground devices.

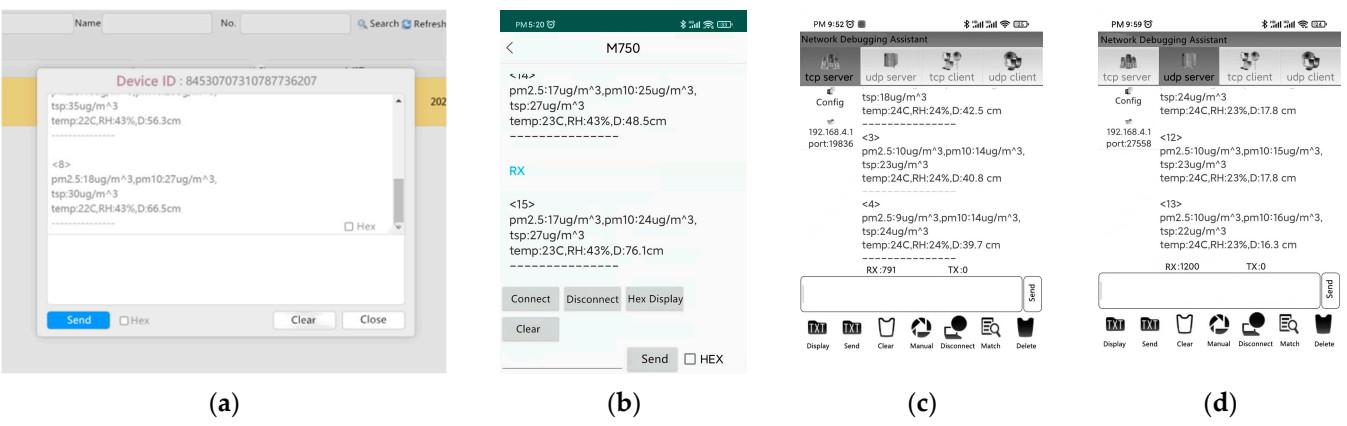

| (**a**) | (**b**) | (**c**) | (**d**) |

**Figure 8.** The monitoring interface of measurement data: (**a**) receiving data through browser in CLOUD mode; (**b**) developed app on Android platform for CLOUD mode; (**c**) TCP mode; (**d**) UPD mode.

### 3.6. Integration and Assembly

To reduce downwash effects from the propellers, we installed the multi-sensor fusion subsystem and the long-distance wireless transmission subsystem on the upper part of the flight power and control subsystem. The continuous working time of the drone measurement system is mainly determined by the power battery of FPCS, and the flight measurement can last about 30 min. We set up separate 11.1 V, 2200 mAh lithium batteries for MSFS and LWCS, which can maintain stable measurement and data transmission for at

least 15 h when fully charged. Figure 9 depicts the final assembled monitoring system with a total weight of 3240 g and a height of 470 mm.

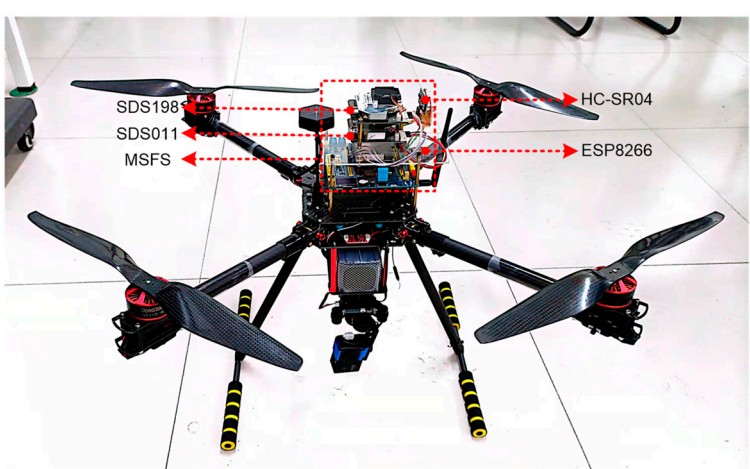

**Figure 9.** Assembled monitoring system.

*3.7. Calibration and Application Experiment*

3.7.1. Sensor Calibration

As mentioned above, both SDS011 and SDS198 have completed data calibration at the factory. However, after the sensor has been used for a period of time, due to the possible contamination of the darkroom and light sensing elements, it is necessary to recalibrate the sensor in addition to necessary cleaning and maintenance. We used two SDS011 (Sensor-A~B) for the calibration experiment of PM2.5 and PM10 and SDS198 (Sensor-C~D) for TSP, where A and C are sensors to be calibrated, and B and D are brand new and reference sensors. To test the sensitivity of the sensor, we adjusted the time resolution to 1 s.

We intercepted 900 consecutive detection points and included the process of artificially adding particle disturbance to these detection points in order to test whether the responses of multiple sensors to the same disturbance are consistent. We calculated the correlation coefficient ($r$) of the two sets of data, carried out linear regression on them, and calculated the coefficient of determination ($R^2$), regression equation, and root mean square error ($RMSE$) after regression. The regression equation is applied to the sensor calibration of the monitoring system when there is a certain deviation after the sensors have worked for a period of time. The equations of correlation coefficient ($r$), coefficient of determination ($R^2$), and root mean square error ($RMSE$) can be described by Equations (9)–(11):

$$r = \frac{\sum\limits_{i=1}^{n} \left(X_i - \overline{X}\right)\left(Y_i - \overline{Y}\right)}{\sqrt{\sum\limits_{i=1}^{n} \left(X_i - \overline{X}\right)^2} \sqrt{\sum\limits_{i=1}^{n} \left(Y_i - \overline{Y}\right)^2}} \tag{9}$$

$$R^2 = \frac{\sum\limits_{i=1}^{n} \left(\hat{Y}_i - \overline{Y}\right)^2}{\sum\limits_{i=1}^{n} \left(Y_i - \overline{Y}\right)^2} \tag{10}$$

$$RMSE = \sqrt{\frac{\sum\limits_{i=1}^{n} \left(\hat{Y}_i - Y_i\right)^2}{n}} \tag{11}$$

where $X_i$ is the data from the sensors that need to be calibrated, $Y_i$ the data from the reference sensors, and $\hat{Y}_i$ is the expected value of the measurement, which can be represented by the regression function $f(X)$:

$$\hat{Y}_i = f(X_i) \tag{12}$$

### 3.7.2. Communication Performance Test

To verify the communication performance of the monitoring system, we conduct 4G and Wi-Fi communication performance experiments. The evaluation indicators of the experiment are mainly communication distance, signal strength, communication accuracy, stable communication time, and so on. The receiving device is a mobile phone(Redmi Note 10 Pro, Beijing, China). Its 4G network signal has 5 gradients from 20 to 100%, and the Wi-Fi signal has 4 gradients from 25 to 100%. We conducted experiments at four distances, 120 m, 150 m, 180 m, and 1500 m. Experiments were conducted in Wenhua Road Campus (37°47′10″ N, 113°39′17″ E) and College of Mechanical and Electrical Engineering (34°47′43″ N, 113°38′56″ E) of Henan Agricultural University, Zhengzhou City, China, as shown in Figure 10.

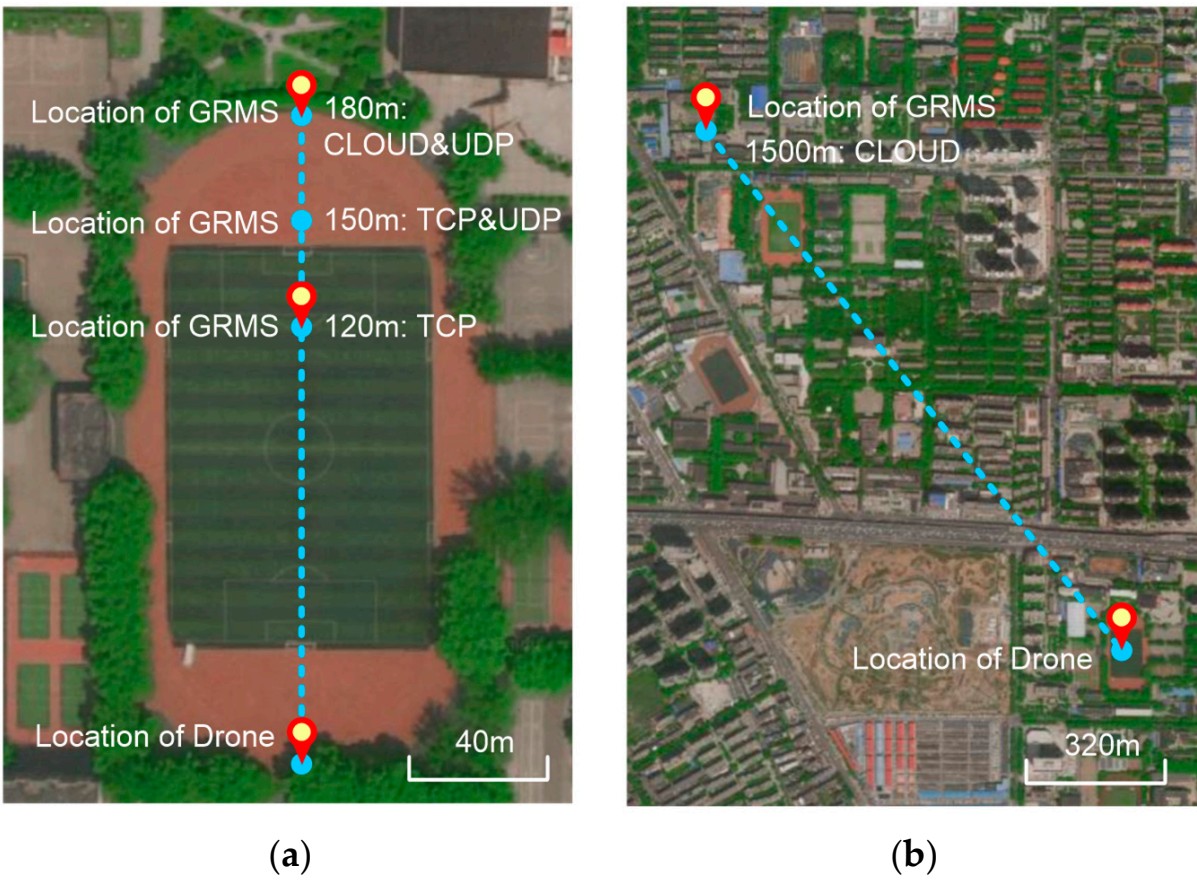

**(a)** **(b)**

**Figure 10.** Four distance settings for communication performance test: (**a**) 120 m, 150 m, and 180 m; (**b**) 1500 m.

### 3.7.3. Flight Measurement

In order to test the performance of the monitoring system and compare the measurement results with the data from the national air quality monitoring stations, we conducted a flight measurement experiment in Wenhua Road Campus from 13:00 to 14:00 on 2 April 2022, as shown in Figure 11.

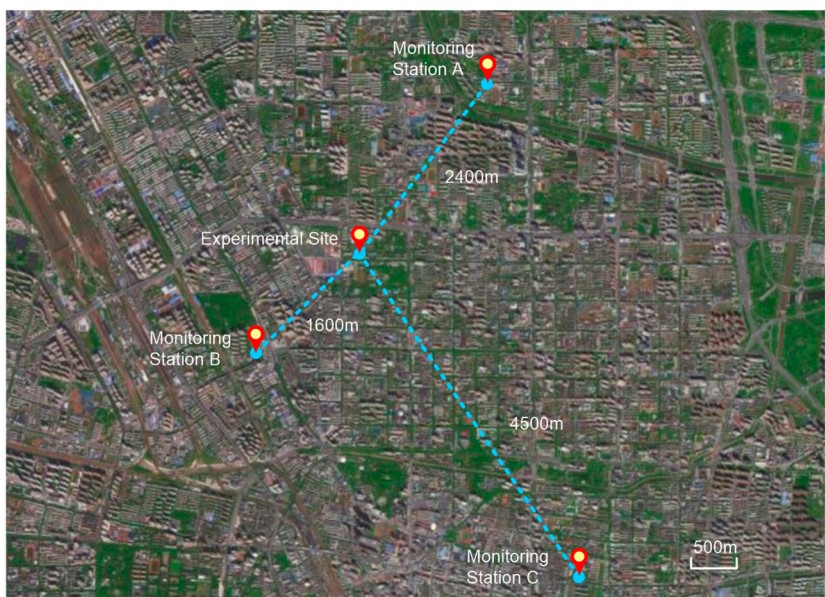

**Figure 11.** Locations of the experimental site for flight measurement. Monitoring stations A, B, and C are the three national monitoring stations closest to the site in different directions.

The ambient temperature is 19.2 °C, the relative humidity is 35%, and the average wind velocity is 1.5–2.2 m/s (SSE). We measured the concentration of particulate matter at three altitudes (5 m, 10 m, and 15 m) at 13:15, 13:30, and 13:45 repeatedly, because the range of 5 to 15 m is the commonly used installation height of the measurement device of the national air quality monitoring stations. The drone monitoring system measured PM concentrations at different altitudes with 1 min hovering. The drone should adopt the bottom-up flight measurement method to obtain the spatial distribution data of particle concentration, because this method can avoid the damage of particle distribution in the area to be measured by the propeller downwash flow. Data were collected every 1 s and averaged over the 1 min of hovering to obtain PM concentrations at each height. The flight measurement experiment is shown in the Figure 12.

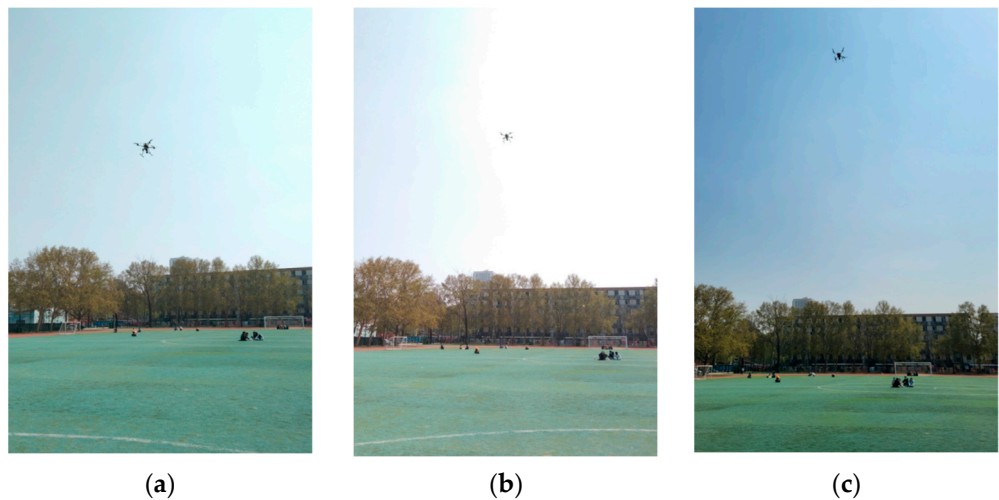

|     (a)     |     (b)     |     (c)     |

**Figure 12.** Flight measurements at different altitudes: (**a**) 5 m; (**b**) 10 m; (**c**) 15 m.

3.7.4. Agri-Environment Measurement

We conducted agri-environmental measurement experiments during wheat harvest in Muzhai Village (34°50′38″ N, 113°22′07″ E), Zhengzhou, China. The experimental site of agri-environment measurement and its surroundings are shown in Figure 13.

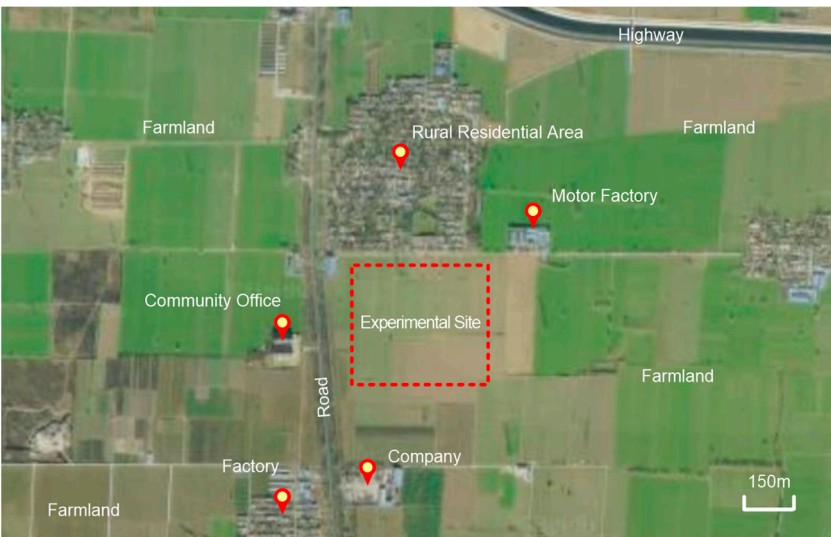

**Figure 13.** Locations of the experimental site for agri-environment measurement.

The experiment was conducted from 8:30 a.m. to 11:30 a.m. on 5 June 2022. The ambient temperature was 27.8–31.1 °C, the relative humidity was 48–39%, and the average wind velocity was 2.4–1.7 m/s (W). During the experiment, a wheat harvester performed reciprocating harvesting operations in a rectangular area of 200 m by 150 m. We set the measurement point in the downwind direction 30 m away from the main activity area of the wheat harvester and measured the PM concentration before and after the harvester started working on that day. The drone monitoring system measured PM concentrations at different altitudes from 0 to 80 m, with 30 s hovering and 10 m vertical rise at each height level. The drone should adopt the bottom-up flight measurement method to obtain the spatial distribution data of particle concentration, because this method can avoid the damage of particle distribution in the area to be measured by the propeller downwash flow. The average value within 30 s of hovering was taken, which is the particle concentration at this height. The power motor of the drone monitoring system was set to the off state at 0 m, and it needed to hover for 1 min at 10 m before measurement. The setting of measurement point and the measurement site are shown in Figure 14.

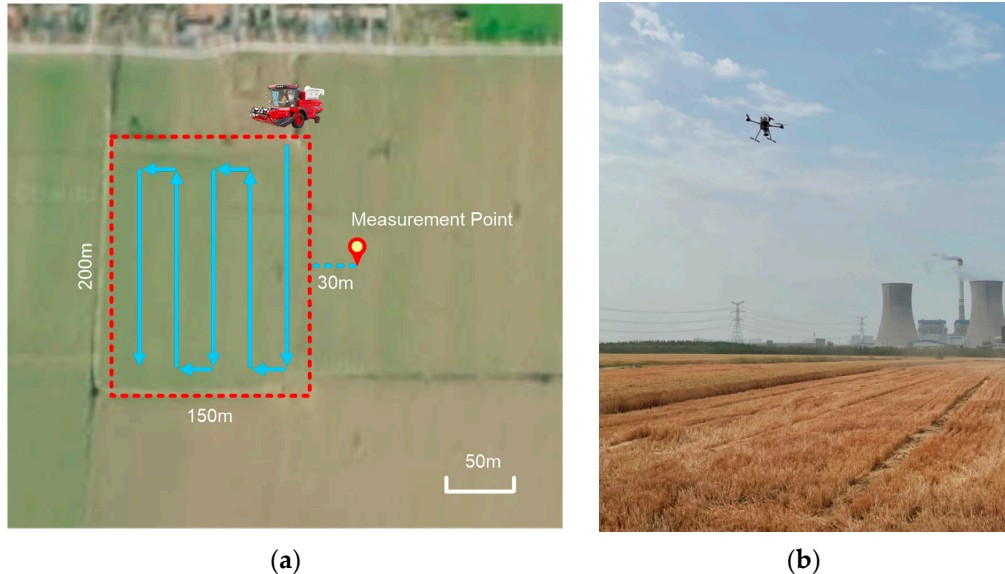

(**a**)                                                                 (**b**)

**Figure 14.** Agri-environmental measurement experimental site: (**a**) setting of measurement point; (**b**) the drone performing flight measurement work.

## 4. Results and Discussion

### 4.1. Sensor Consistency and Calibration

When SDS011 reads the PM2.5 and PM10 measurement values, one decimal place is reserved by default. This section retains this setting for the sake of test accuracy. However, in the final monitoring system, in order to be consistent with the reading method of SDS198 that does not retain decimals, we used integers to read the concentrations. The measurement and processing results of sensors at 900 time points are shown in Figure 15.

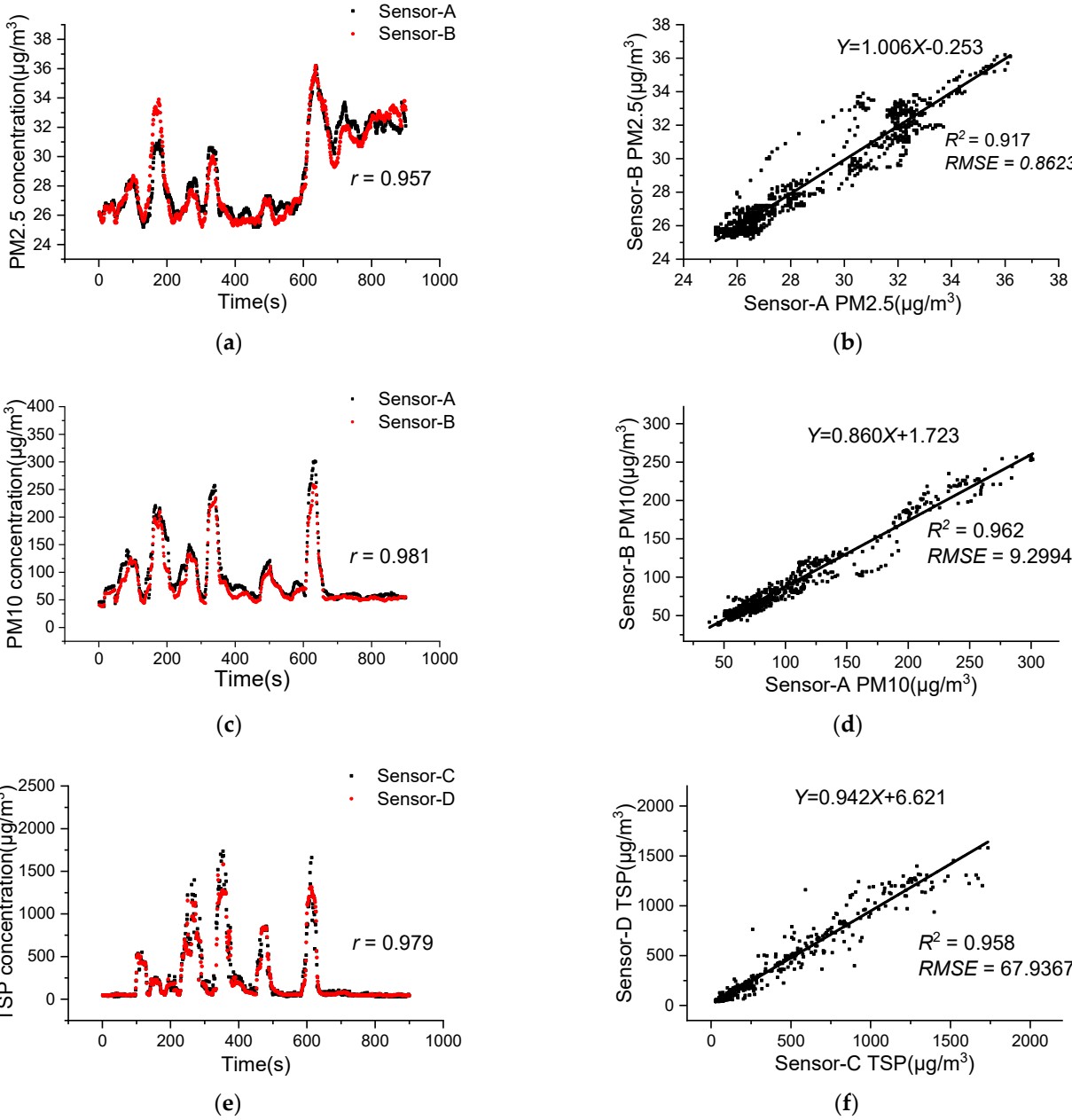

**Figure 15.** Sensor consistency and calibration results: (**a**) PM2.5 measurement consistency; (**b**) Sensor-A PM2.5 measurement calibration; (**c**) PM10 measurement consistency; (**d**) Sensor-A PM10 measurement calibration; (**e**) TSP measurement consistency; (**f**) Sensor-C TSP measurement calibration.

It can be seen from Figure 15a,c that the although there is some deviation in the measurement value between the Sensor-A that has been used for a period of time and the new Sensor-B, the responses of the two sensors to the artificial disturbance of particulate

matter are very consistent. A similar relationship exists between Sensor-C and -D in Figure 15e. The correlation of PM concentration between sensors is very high, and the correlation coefficients of PM2.5, PM10, and TSP reach 0.957, 0.981, and 0.979, respectively.

After statistics, the average concentration values of PM2.5, PM10, and TSP measured by Sensor-A and -C, which have been used for a period of time, are 0.26%, 13.79%, and 3.30% higher than those of the new Sensor-B and -D, respectively. This may be due to long-term exposure to particulate matter, resulting in contamination of the optical sensor. Figure 15b,d,f show the results of calibration of the sensor, and the coefficient of determination of the three kinds of particle concentrations reach 0.917, 0.962, and 0.958 respectively.

### 4.2. Communication Suitability

The results of the communication performance test are shown in Table 3.

**Table 3.** Performance test results of three communication modes.

| Parameter | Distance | Signal Strength | Communication Accuracy | Max Duration |
|---|---|---|---|---|
| 4G-CLOUD | 180 m | 100% | 99.38% | >40 min |
|  | 1500 m | 100% | 99.63% | >40 min |
| AP-TCP | 120 m | 75% | 100% | >40 min |
|  | 150 m | 50% | 100% | 23.7 min |
| AP-UDP | 150 m | 50% | 97.13% | >40 min |
|  | 180 m | 25% | 93.88% | >40 min |

The test results demonstrated that the signal strength, communication accuracy, and stable connection duration of the 4G-CLOUD mode are not limited by the communication distance and have excellent communication performance, which should be used as the preferred communication mode. The 4G-CLOUD mode uses a 4G network for communication, so it is not restricted by obstacles and can realize ultra-long-distance communication. Since it transfers data through a cloud server, multiple devices can receive data at the same time, but it will fail in areas without 4G network signals.

If there is no 4G network signal, when the distance is less than 120 m, the AP-TCP mode can provide very stable and reliable continuous data transmission. When the distance is between 120 and 180 m, the connection interruption will occur in the AP-TCP mode. It is more suitable to use the AP-UDP mode at this time, while its communication accuracy will decrease as the distance increases. When the distance exceeds 180 m, the Wi-Fi communication method is no longer reliable. The AP-TCP mode needs to keep the connection smooth at all times. When the signal is poor, there will be transmission congestion or connection interruption. It needs to wait or reconnect to transmit data. When the signal is strong, this mode has higher reliability and real-time performance, which is mainly used for close-range and barrier-free data communication. Packet loss may occur when transmitting data in AP-UDP mode, but this mode does not require reconnection, and subsequent packet transmission will not be affected after packet loss. When the transmission distance is moderate and the signal strength cannot be fully guaranteed, there will be certain advantages.

### 4.3. Flight Measurement and Comparison

The results of the flight measurement experiment on 2 April 2022 and the PM concentration measurement data of the three national monitoring stations at the same time are shown in Table 4.

**Table 4.** Flight measurement data at three altitudes and data from national monitoring stations.

| Parameter | 5 m | 10 m | 15 m | AVG | Station A | Station B | Station C |
|-----------|-----|------|------|-----|-----------|-----------|-----------|
| PM2.5 ($\mu g/m^3$) | 38 | 39 | 42 | 39.7 | 36 | 38 | 39 |
| PM10 ($\mu g/m^3$) | 67 | 73 | 72 | 70.7 | 61 | 73 | 71 |

As TSP is not the monitoring object of the national monitoring stations, there are no relevant data about TSP. Therefore, only the mass concentrations of PM2.5 and PM10 are listed in the table. In order to compare with the data of the national monitoring station, we used the inverse distance weight (IDW) interpolation method to calculate the spatial interpolation of particle concentration in the experimental site according to the data from stations A, B, and C. The solution of inverse distance weight interpolation can be described by the following equation:

$$w_i = \frac{\frac{1}{d_i}}{\sum\limits_{i=1}^{n} \frac{1}{d_i}} \quad (13)$$

$$\hat{C}_E = \sum\limits_{i=1}^{n} w_i C_i \quad (14)$$

where $w_i$ is the weight of station $i$ when calculating spatial interpolation, which is related to the reciprocal of the distance $d_i$ from station $i$ to the experimental site. $\hat{C}_E$ is the spatial interpolation of the PM concentration at the experimental site, and $C_i$ is the monitoring value at station $i$. The calculation results are shown in Figure 16.

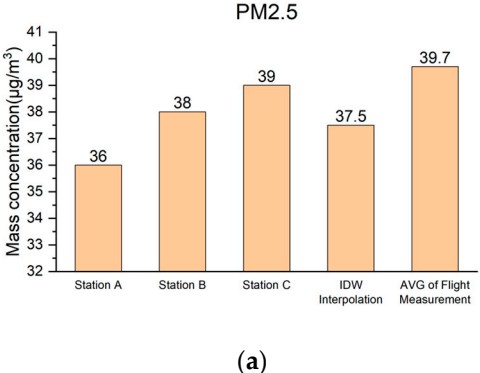

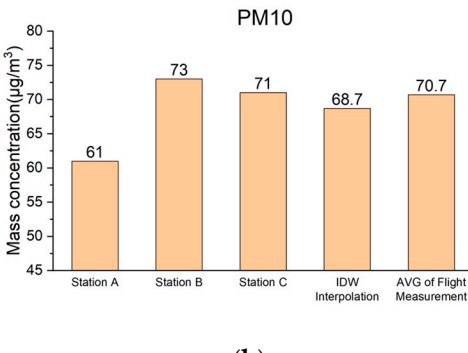

(**a**)  (**b**)

**Figure 16.** Comparison of spatial interpolation of PM concentration with flight measurement: (**a**) PM2.5; (**b**) PM10.

From the experimental results, it can be seen that the flight measurements of PM2.5 and PM10 are very close to the calculated spatial interpolation, and the errors are only 5.87% and 2.91%, respectively. In the three monitoring stations, although station C is the farthest from the experimental site, the PM concentration is the closest to the flight measurement value, presumably because station C is in the upwind direction of the experimental site, and the correlation between them is greater. The experimental results shows that our environmental monitoring system has high application value.

*4.4. Agri-Environment Measurement Experiment Results*

We took the particle concentration measured before the harvester started working as the initial concentration. The difference between the concentration measured at the same height during the operation of the harvester and the initial concentration can be considered as the particle emission from the harvester. Figure 17 depicts the measurement results of PM concentration at different altitudes.

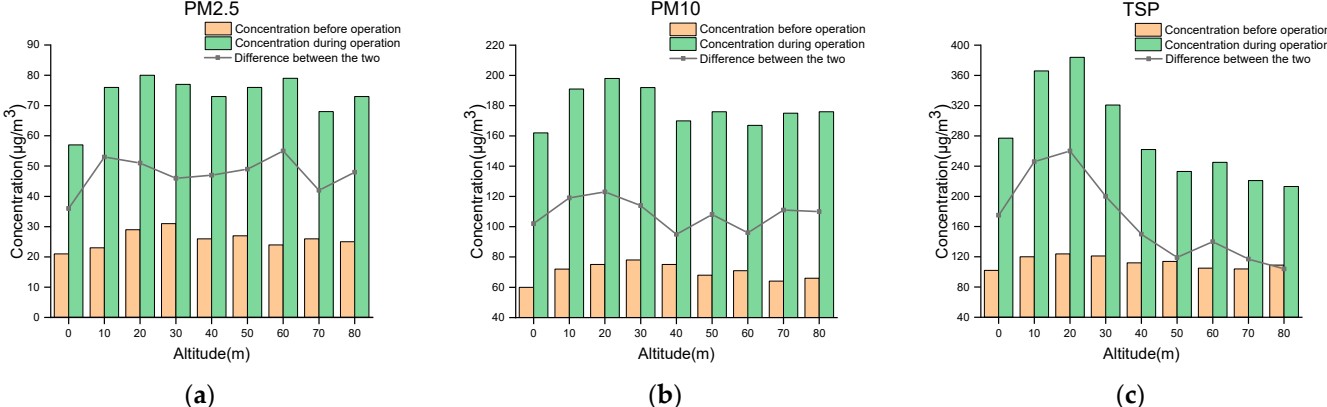

**Figure 17.** Comparison of PM concentration at measurement point before and after operation of harvester started working: (**a**) PM2.5; (**b**) PM10; (**c**) TSP.

From the experimental results, it can be seen that before the harvester started working, the concentration of three types of PM followed a unimodal pattern, starting from the minimum concentration at ground level, reaching the maximum at 30 m or 20 m, and then fluctuating and slowly declining with increasing altitude. The concentration of TSP decreased faster than the other two kinds of particles [29,30], which may be because the particles of 10~100 μm settle faster due to their larger mass [31,32]. After the harvester started working for a period of time, the concentrations of three types of PM all increased significantly, and their changing trends with increasing altitude were also different from those before. Although PM2.5 reached the maximum value at 20 m, it was not outstanding compared with the values at other altitudes. Except that the concentration at the ground was still the lowest, there was no obvious change trend with increasing altitude. The peak of PM10 appeared at the altitude of 20 m, which is lower than before. With increasing altitude, the concentration decreased rapidly between 20–40 m, and then became stable. The peak of TSP is obvious, which still appeared at the altitude of 20 m, and the concentration decreased more rapidly between 20–40 m than 40–80 m.

The curve "PM from harvester" in the figure could be understood as the concentration of particulate matter emitted by the harvester into the surrounding environment during operation, and the curve had a very obvious peak at the altitude of 10–20 m. This suggested that particulate emissions from harvesters may be concentrated at this altitude. Due to the different altitudes of the concentrated emission of particulate matter, the impact on the surrounding environment is also different. For example, the higher the altitude, the easier it is for the particulate matter to spread to a farther place under the action of wind. Therefore, this result is important for the study of particulate matter emissions in agricultural environments and their effects. At the same time, the altitude of 15 m can be used as the best measurement altitude for the particle emission of the harvester. This altitude may vary depending on the location, angle, and wind speed of the harvester's outlet.

## 5. Conclusions

We developed a new platform for agri-environment atmospheric monitoring using a highly maneuverable drone equipped with multi-sensor and long-distance communication systems. This platform has the characteristics of small size, lightweight, and high cost-effectiveness, and also high temporal and spatial resolution. We tested the sensor consistency and proposed a calibration method. The light scattering sensor had a high consistency for PM2.5, PM10, and TSP measurement even after a period of use, and the correlation coefficients were all above 0.95. In the area covered by 4G signal, the communication accuracy of the 4G-CLOUD communication mode reached more than 99% and was not affected by distance and obstacles, so it could be used as the preferred communication mode. If there was no 4G signal, the reliable communication distance of the monitoring

platform was 180 m. The AP-TCP mode could provide stable data transmission in a short distance, but when the distance exceeded 120 m, AP-UDP was required for communication. We conducted a flight measurement comparison experiment, and the measured PM2.5 and PM10 concentrations were very close to the data from the national monitoring station. We carried out the agri-environment atmospheric measurement experiment and found that the PM concentrations at the measuring point were significantly different before and after the harvester started working, and the location of the peak changed, which is very significant for the selection of measurement altitude in the future related research.

Compared with national monitoring stations, the biggest advantage of our mobile monitoring system is that it has the ability to measure the concentration of TSP and the vertical distribution of PM, which is very important for the research of agricultural environmental particulate matter emission characteristics. Unlike national monitoring stations that can only measure PM2.5 and PM10, the measurement function of TSP considers the particularity of the agri-environment, which makes the spatiotemporal distribution data of the agri-environment atmospheric measurement more substantial.

In future work, we should give more consideration to the distribution patterns of particles in height under various conditions (such as pressure, wind speed, humidity, etc.), and use fixed measuring devices at different measuring positions to obtain the horizontal distribution characteristics of near-ground particulate matter, so as to provide more agricultural environmental atmospheric data for the relevant research on agricultural particulate matter emission reduction.

**Author Contributions:** Conceptualization, Y.L. and W.W.; methodology, X.H.; software, C.Z.; validation, Y.L. and W.W.; formal analysis, Y.L.; investigation, X.H. and R.J.; writing—original draft preparation, Y.L. and C.Z.; writing—review and editing, Y.L. and W.W.; visualization, R.J. and J.C.; funding acquisition, X.H. and W.W. All authors have read and agreed to the published version of the manuscript.

**Funding:** This research was funded by the China Agriculture Research System (CARS-03) and Major Public Research Projects in Henan Province (201300110400).

**Institutional Review Board Statement:** Not applicable.

**Data Availability Statement:** The data used to support the findings of this study are available from the corresponding author upon request.

**Conflicts of Interest:** The authors declare no conflict of interest.

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
