# Peer review of "Agri-Environment Atmospheric Real-Time Monitoring Technology Based on Drone and Light Scattering"

_agriculture, doi:10.3390/agriculture12111885_

Round 1

Reviewer 1 Report

The authors have presented a manuscript, describing the agri environment atmospheric real-time monitoring technology based on drone and light scattering. Following, I have included some comments to improve the manuscript.

  1. I suggest to the authors to add a new section detailing the state of the art. In this section, authors have to describe the relevant related work in which explain.
  2. Can the authors include at the end of the introduction, more details of the objectives of their study.
  3. This work presents very interesting results and practice to response of monitoring technology based on drone and light scattering. I think that the authors can improve the format of results demonstration. The authors can highlight better the importance of the results obtained.
  4. Conclusions. Consider extending the conclusions and adding a Future works paragraph. The summary and Conclusions, it is better to combine them in only section of Conclusions.
  5. References: Line 554 and 573. Author must also present the change the year to bold and put the reference page.

Finally, the review is interesting and presents considerable information on the possibilities of the agri environment atmospheric real time monitoring technology based on drone and light scattering, but authors must improve the presentation of their results and discussion. The topic in interesting, but the study lacks more details with precision, and concrete conclusion that will help farmers to improve or to change their strategies in the agriculture.

Reviewer 2 Report

The idea presented by the authors seems very interesting. Some of the suggestions from my side can be considered:

The illustration in figure 1 is missing

On lines 106-107, the authors claimed that “Our system has a long-distance communication function, which can transmit the measurement data to the receiving device on the ground for real-time monitoring.” How the long-distance communication has been achieved and how this method is superior to already existing schemes?

Change the background of figures 9c and 9d to white.

Try to cite equations 9-11.

The following article can be useful for the authors to help enrich the content of the current version.

Implementation of a LoRaWAN Based Smart Agriculture Decision Support System for Optimum Crop Yield” Sustainability 2022, Vol. 14, Issue 2, 827. DOI: 10.3390/su14020827.

Reviewer 3 Report

Dear Authors.

The article describes up-to date topic like the application of drones in agriculture. But, as the problem needs further enhancements, it should be stated, what further works should be done in the future. Chapter Materials and methods contains a lot  information, is very long, and it should be re-considered if some information need to be moved to Results part (f.e. figure 7), taking into account, that huge part of work done it this article is to set up the measuring equipment. On the other hand: subchapter 3.1- equations should be moved to chapter 2nd Materials and methods . The abstract, discussion, literature needs improvements.

I have also following detailed remarks:

1) In abstract: lack of information about results obtained regarding level of emissions. From the reader’s point of view the most valuable are data obtained from measurements.

2) L. 50 soil erosion by wind is not agricultural activity, but- it is factor.

3) What is the main aim the study?  Is it only to produce the high quality sensors, like stated in L. 103-L.106? Add adequate information to abstract and main text.

4) Almost no discussion, comparing other results from such types of measurements, using drones. Indicate the novelty of article, in the main text in part Discussion and in abstract.

5) in table 2- what exactly “max value” mean?

6) figure 9- make better quality of images, now there are not visible enough.

7) figure 10- should be added markings links on the figure, f.e to show where particular sensors and MSFS are placed.

8) subchapter 2.7. –rename it, because “methods” are included in the name of main Chapter 2.

9) for how long time each day the measurements could be performed? Have there been trials with longer measurement times? Was the system stable in long-term conditions? For the monitoring purposes, multi-hour measurements are important from the end user's point of view.

10) In conclusions should be stated, what are the biggest advantages of a mobile monitoring system (drones) over a fixed system (state’s system)?

11) 10 out of 31 positions of literature is older than 7 years, in some cases it really could be replaced by the newer one.

Round 2

Reviewer 2 Report

The authors have addressed my comments positively and now the article seems ready for publication in its present form. Thank You.

Reviewer 3 Report

Dear Authors.

 All improvements were made, but additionally, the quality of descriptions in the legend in figure 17 is wrong.

Generally, I do not have further remarks.